

# Spectral representation of Matsubara n-point functions: Exact kernel functions and applications

**Johannes Halbinger⋆, Benedikt Schneider and Björn Sbierski**

Department of Physics and Arnold Sommerfeld Center for Theoretical Physics (ASC),
Ludwig-Maximilians-Universität München, Theresienstr. 37, München D-80333, Germany
Munich Center for Quantum Science and Technology (MCQST),
Schellingstr. 4, D-80799 München, Germany

⋆ johannes.halbinger@physik.uni-muenchen.de

## Abstract

In the field of quantum many-body physics, the spectral (or Lehmann) representation simplifies the calculation of Matsubara $n$-point correlation functions if the eigensystem of a Hamiltonian is known. It is expressed via a universal kernel function and a system- and correlator-specific product of matrix elements. Here we provide the kernel functions in full generality, for arbitrary $n$, arbitrary combinations of bosonic or fermionic operators and an arbitrary number of anomalous terms. As an application, we consider bosonic 3- and 4-point correlation functions for the fermionic Hubbard atom and a free spin of length $S$, respectively.

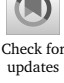

# 1 Introduction

Multi-point correlation functions of $n$ quantum mechanical operators, also known as $n$-point functions, are a central concept in the study of quantum many-body systems and field theory [1]. They generalize the well-known 2-point functions, which, for the example of electrons in the solid state, are routinely measured by scanning tunneling spectroscopy or angle-resolved photon emission spectroscopy [2]. For magnetic systems, the 2-point spin correlators can be probed in a neutron scattering experiment. Higher order correlation functions with $n = 3, 4, 5...$ can for example be measured in non-linear response settings [3]. In the emerging field of cold atomic quantum simulation, (equal-time) $n$-point functions are even directly accessible [4].

On the theoretical side the study of higher order correlation functions gains traction as well. One motivation is the existence of exact relations between correlation functions of different order $n$ [5, 6]. Although these exact relations can usually not be solved exactly, they form a valuable starting point for further methodological developments like the parquet approximation [7]. Thus even if the 4-point correlator (or, in that context, its essential part, the one-line irreducible vertex [1]) might not be the primary quantity of interest in a calculation, it appears as a building block of the method. Another example is the functional renormalization group method (fRG) in a vertex expansion [8, 9]. It expresses the many body problem as a hierarchy of differential equations for the vertices that interpolate between a simple solvable starting point and the full physical theory [10]. Whereas experiments measure correlation functions in real time (or frequency), in theory one often is concerned with the related but conceptually simpler versions depending on imaginary time [1]. In the following, we will focus on these Matsubara correlation functions, which, nevertheless feature an intricate frequency dependence.

Whereas the above theoretical methods usually provide only an approximation for the $n$-point functions, an important task is to calculate these objects exactly. This should be possible for simple quantum many body systems. We consider systems simple if they are amenable to exact diagonalization (ED), i.e. feature a small enough Hilbert space, like few-site clusters of interacting quantum spins or fermions. Also impurity systems, where interactions only act locally, can be approximately diagonalized using the numerical renormalization group [11].

Knowing the exact $n$-point functions for simple systems is important for benchmark testing newly developed methods before deploying them to harder problems. Moreover, $n$-point functions for simple systems often serve as the starting point of further approximations like in the spin-fRG [12–14], or appear intrinsically in a method like in diagrammatic extensions of dynamical mean field theory [15] with its auxiliary impurity problems. Another pursuit enabled by the availability of exact $n$-point functions is to interpret the wealth of information encoded in these objects, in particular in their rich frequency structure. For example, Ref. [16] studied the fingerprints of local moment formation and Kondo screening in quantum impurity models.

In this work we complete the task to calculate exact $n$-point functions by generalizing the spectral (or Lehmann) representation [1, 17] for Matsubara $n$-point correlation functions to arbitrary $n$. We assume that a set of eigenstates and -energies is given. Following pioneering work of Refs. [18–20] and in particular the recent approach by Kugler *et al.* [21], we split the problem of calculating imaginary frequency correlators into the computation of a universal kernel function and a system- and correlator-specific part (called partial spectral function in Ref. [21]). We provide the kernel functions in full generality for an arbitrary number $n$ of bosonic or fermionic frequencies. Previously, these kernel functions were known exactly only up to the 3-point case [18], for the fermionic 4-point case [19–21] or for the general $n$-point case [21] but disregarding anomalous contributions to the sum that the kernel function con-

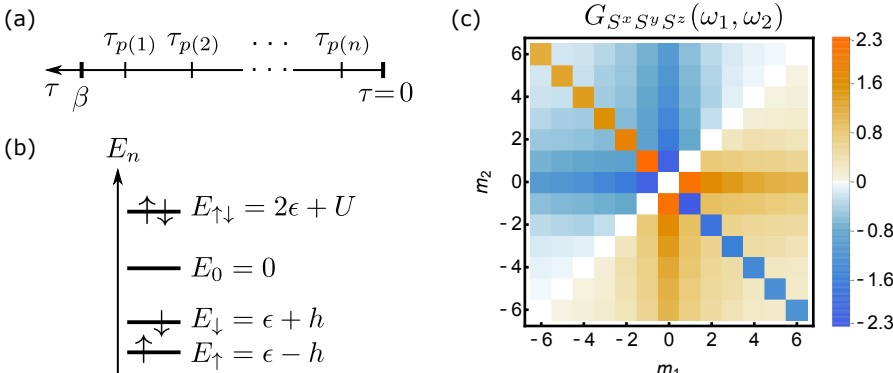

Figure 1: (a) Ordering convention for imaginary times in Eq. (9). (b) Eigenstates and energies of the Hubbard atom. (c) Matsubara correlation function $G_{S^x S^y S^z}(\omega_1, \omega_2)$ with $\omega_j = 2\pi m_j/\beta$ ($m_j \in \mathbb{Z}$, $j = 1, 2$) for the Hubbard atom (35) at $\beta = 10$, $h = 0.1$, $\epsilon = -2$, $U = 2$, see Eq. (45). The sharp anti-diagonal ray $\propto \delta_{\omega_1 + \omega_2, 0}$ represents an anomalous term of order $a = 1$. The other broadened rays become sharp and anomalous for $h \to 0$, see Eq. (49).

sists of. These anomalous contributions are at the heart of the complexity of Matsubara $n$-point functions. They occur when certain combinations of eigenenergies and external frequencies vanish individually, see the anti-diagonal rays in Fig. 1(c). Physically, they correspond to long-term memory effects, are related to non-ergodicity and, in the case of bosonic two-point functions reflect the difference between static isothermal susceptibilities and the zero-frequency limit of the dynamical Kubo response function [22, 23].

The structure of the paper is as follows: In Sec. 2 we define the Matsubara $n$-point function $G_{A_1...A_n}(\omega_1, ..., \omega_{n-1})$ and review some of its properties. The spectral representation is derived in Sec. 3 with Eq. (15) being the central equation written in terms of the kernel function $K_n(\Omega_1, ..., \Omega_{n-1})$. Our main result is an exact closed-form expression of this most general kernel function which is given in Sec. 4. Examples for $n = 2, 3, 4, 5$ are given in Sec. 5 where we also discuss simplifications for the purely fermionic case. We continue with applications to two particular systems relevant in the field of condensed matter theory: In Sec. 6, we consider the Hubbard atom and the free spin of length $S$, for which we compute $n$-point functions not previously available in the literature. We conclude in Sec. 7.

## 2 Definition of Matsubara $n$-point function $G_{A_1...A_n}(\omega_1, ..., \omega_n)$

We consider a set of $n = 2, 3, 4, ...$ operators $\{A_1, A_2, ..., A_n\}$ defined on the Hilbert space of a quantum many-body Hamiltonian $H$. The operators can be fermionic, bosonic or a combination of both types, with the restriction that there is an even number of fermionic operators. As an example, $A_1 = d^\dagger d \equiv n$, $A_2 = d$, $A_3 = d^\dagger$ where $d^\dagger$ and $d$ are canonical fermionic creation and annihilation operators. A subset of operators is called bosonic if they create a closed algebra under the commutation operation. They are called fermionic if the algebra is closed under anti-commutation, see Sec. 1 of Ref. [24]. Spin operators are thus bosonic.

We define the imaginary time-ordered $n$-point correlation functions for imaginary times $\tau_k \in [0, \beta]$, [25, 26],

$$G_{A_1 A_2...A_n}(\tau_1, \tau_2, ..., \tau_n) \equiv \langle \mathcal{T} A_1(\tau_1) A_2(\tau_2)..A_n(\tau_n) \rangle, \qquad (1)$$

where $A_k(\tau_k) = e^{\tau_k H} A_k e^{-\tau_k H}$ denotes Heisenberg time evolution. Here and in the following, $k = 1, 2, ..., n$. The expectation value is calculated as $\langle ... \rangle = \text{tr}[\rho ...]$ where $\rho = \exp(-\beta H)/Z$

is the thermal density operator at inverse temperature $\beta = 1/T$ and $Z = \operatorname{tr} \exp(-\beta H)$ is the partition function. Note that other conventions for the $n$-point function differing by a prefactor are also used in the literature, e.g. Ref. [21] multiplies with $(-1)^{n-1}$. In Eq. (1), the imaginary time-ordering operator $\mathcal{T}$ orders the string of Heisenberg operators,

$$\mathcal{T}A_1(\tau_1)A_2(\tau_2)...A_n(\tau_n) \equiv \zeta(p)A_{p(1)}\big(\tau_{p(1)}\big)A_{p(2)}\big(\tau_{p(2)}\big)...A_{p(n)}\big(\tau_{p(n)}\big), \qquad (2)$$

where $p$ is the permutation $p \in S_n$ such that $\tau_{p(1)} > \tau_{p(2)} > ... > \tau_{p(n)}$ [see Fig. 1(a)] and the sign $\zeta(p)$ is $-1$ if the operator string $A_{p(1)}A_{p(2)}...A_{p(n)}$ differs from $A_1 A_2...A_n$ by an odd number of transpositions of fermionic operators, otherwise it is $+1$. The special case $n = 2$, with $\zeta(12) = 1$ and $\zeta(21) = \zeta$ ($\zeta = 1$ for $A_{1,2}$ bosonic, $\zeta = -1$ for $A_{1,2}$ fermionic), simplifies to

$$\mathcal{T}A_1(\tau_1)A_2(\tau_2) = \begin{cases} A_1(\tau_1)A_2(\tau_2), & \tau_1 > \tau_2, \\ \zeta A_2(\tau_2)A_1(\tau_1), & \tau_2 > \tau_1. \end{cases} \qquad (3)$$

Imaginary time-ordered correlation functions (1) fulfill certain properties which we review in the following, see e.g. [26] for a more extensive discussion. First, they are invariant under translation of all time arguments,

$$G_{A_1 A_2...A_n}(\tau_1, \tau_2, ..., \tau_n) = G_{A_1 A_2...A_n}(\tau_1 + \tau, \tau_2 + \tau, ..., \tau_n + \tau), \qquad (4)$$

with $\tau \in \mathbb{R}$ such that $\tau_k + \tau \in [0, \beta]$. They also fulfill periodic or anti-periodic boundary conditions for the individual arguments $\tau_k$,

$$G_{A_1...A_n}(\tau_1, ..., \tau_k = 0, ..., \tau_n) = \zeta_k G_{A_1...A_n}(\tau_1, ..., \tau_k = \beta, ..., \tau_n), \qquad (5)$$

where $\zeta_k = +1$ or $-1$ if $A_k$ is from the bosonic or fermionic subset of operators, respectively. This motivates the use of a Fourier transformation,

$$G_{A_1...A_n}(\tau_1, ..., \tau_n) \equiv \beta^{-n} \sum_{\omega_1,...,\omega_n} e^{-i(\omega_1\tau_1+...+\omega_n\tau_n)} G_{A_1...A_n}(\omega_1, ..., \omega_n), \qquad (6)$$

$$G_{A_1...A_n}(\omega_1, ..., \omega_n) = \int_0^\beta d\tau_1 \cdots \int_0^\beta d\tau_n e^{+i(\omega_1\tau_1+...+\omega_n\tau_n)} G_{A_1...A_n}(\tau_1, ..., \tau_n), \qquad (7)$$

where $\omega_k = 2\pi m_k/\beta$ or $\omega_k = 2\pi(m_k + 1/2)/\beta$ with $m_k \in \mathbb{Z}$ are bosonic or fermionic Matsubara frequencies, respectively, and $\sum_{\omega_k}$ is shorthand for $\sum_{m_k \in \mathbb{Z}}$. Note that fermionic Matsubara frequencies are necessarily nonzero, a property that will become important later. As we will not discuss the real-frequency formalisms, we will not write the imaginary unit in front of Matsubara frequencies in the arguments of $G_{A_1...A_n}(\omega_1, ..., \omega_n)$. Again, note that in the literature, different conventions for the Fourier transformation of $n$-point functions are in use. In particular some authors pick different signs in the exponent of Eq. (7) for fermionic creation and annihilation operators, or chose these signs depending on operator positions.

Time translational invariance (4) implies frequency conservation at the left hand side of Eq. (7),

$$G_{A_1...A_n}(\omega_1, ..., \omega_{n-1}, \omega_n) \equiv \beta \delta_{0,\omega_1+...+\omega_n} G_{A_1...A_n}(\omega_1, ..., \omega_{n-1}), \qquad (8)$$

where on the right hand side we skipped the $n$-th frequency entry in the argument list of $G$. Note that we do not use a new symbol for the correlation function when we pull out the factor $\beta$ and the Kronecker delta function.

# 3 Spectral representation of $G_{A_1...A_n}(\omega_1, ..., \omega_{n-1})$

The integrals involved in the Fourier transformation (7) generate all $n!$ different orderings of the time arguments $\tau_k$. As in Ref. [21] it is thus convenient to use a sum over all $n!$ permutations $p \in S_n$ and employ a product of $n-1$ step-functions $\theta$, with $\theta(x) = 1$ for $x > 0$ and $0$ otherwise, to filter out the unique ordering for which $\beta > \tau_{p(1)} > \tau_{p(2)} > ... > \tau_{p(n-1)} > \tau_{p(n)} > 0$, see Fig. 1(a),

$$G_{A_1...A_n}(\tau_1, ..., \tau_n) = \sum_{p \in S_n} \zeta(p) \left[ \prod_{i=1}^{n-1} \theta(\tau_{p(i)} - \tau_{p(i+1)}) \right] \langle A_{p(1)}(\tau_{p(1)}) A_{p(2)}(\tau_{p(2)}) ... A_{p(n)}(\tau_{p(n)}) \rangle . \quad (9)$$

To expose explicitly the time dependence of the Heisenberg operators, we insert $n$ times the basis of eigenstates and -energies of the many-body Hamiltonian $H$. Instead of the familiar notation $|j_1\rangle, |j_2\rangle, ...$ and $E_{j_1}, E_{j_2}, ...$ we employ $|\underline{1}\rangle, |\underline{2}\rangle, ...$ and $E_{\underline{1}}, E_{\underline{2}}, ...$ for compressed notation and denote operator matrix elements as $A^{\underline{12}} = \langle \underline{1}|A|\underline{2}\rangle$. We obtain

$$G_{A_1...A_n}(\tau_1, ..., \tau_n) = \sum_{p \in S_n} \zeta(p) \left[ \prod_{i=1}^{n-1} \theta(\tau_{p(i)} - \tau_{p(i+1)}) \right] \quad (10)$$

$$\times \frac{1}{Z} \sum_{\underline{1}...\underline{n}} e^{-\beta E_{\underline{1}}} e^{\tau_{p(1)} E_{\underline{1}}} A_{p(1)}^{\underline{12}} e^{(-\tau_{p(1)} + \tau_{p(2)}) E_{\underline{2}}} A_{p(2)}^{\underline{23}} e^{(-\tau_{p(2)} + \tau_{p(3)}) E_{\underline{3}}} ... e^{(-\tau_{p(n-1)} + \tau_{p(n)}) E_{\underline{n}}} A_{p(n)}^{\underline{n1}} e^{-\tau_{p(n)} E_{\underline{1}}},$$

and apply the Fourier transform according to the definition (7),

$$G_{A_1...A_n}(\omega_1, ..., \omega_n) = \frac{1}{Z} \sum_{p \in S_n} \zeta(p) \sum_{\underline{1}...\underline{n}} e^{-\beta E_{\underline{1}}} A_{p(1)}^{\underline{12}} A_{p(2)}^{\underline{23}} ... A_{p(n)}^{\underline{n1}} \quad (11)$$

$$\times \left[ \int_0^\beta d\tau_{p(1)} e^{\Omega_{p(1)}^{\underline{12}} \tau_{p(1)}} \right] \left[ \int_0^{\tau_{p(1)}} d\tau_{p(2)} e^{\Omega_{p(2)}^{\underline{23}} \tau_{p(2)}} \right]$$

$$\times \cdots \times \left[ \int_0^{\tau_{p(n-2)}} d\tau_{p(n-1)} e^{\Omega_{p(n-1)}^{\underline{n-1\,n}} \tau_{p(n-1)}} \right] \left[ \int_0^{\tau_{p(n-1)}} d\tau_{p(n)} e^{\Omega_{p(n)}^{\underline{n1}} \tau_{p(n)}} \right],$$

where we defined

$$\Omega_k^{\underline{a}\,\underline{b}} \equiv i\omega_k + E_{\underline{a}} - E_{\underline{b}} \in \mathbb{C} . \quad (12)$$

In Eq. (11), the first line carries all the information of the system and the set of operators $\{A_1, A_2, ..., A_n\}$. The remaining terms can be regarded as a universal kernel function defined for general $\{\Omega_1, \Omega_2, ..., \Omega_n\}$ probed at $\Omega_k \in \mathbb{C}$ which depends on the system and correlators via (12). Upon renaming the $\tau$-integration variables $\tau_{p(k)} \to \tau_k$, this kernel function is written as follows:

$$\mathcal{K}_n(\Omega_1, ..., \Omega_n) \equiv \left[ \int_0^\beta d\tau_1 e^{\Omega_1 \tau_1} \right] \left[ \int_0^{\tau_1} d\tau_2 e^{\Omega_2 \tau_2} \right] ... \left[ \int_0^{\tau_{n-2}} d\tau_{n-1} e^{\Omega_{n-1} \tau_{n-1}} \right] \left[ \int_0^{\tau_{n-1}} d\tau_n e^{\Omega_n \tau_n} \right] \quad (13)$$

$$\equiv \beta \delta_{0, \Omega_1 + \Omega_2 + ... + \Omega_n} K_n(\Omega_1, ..., \Omega_{n-1}) + R_n(\Omega_1, ..., \Omega_n) . \quad (14)$$

In the second line we split $\mathcal{K}_n$ into a part $K_n$ proportional to $\beta \delta_{0, \Omega_1 + \Omega_2 + ... + \Omega_n}$ and the rest $R_n$. We dropped $\Omega_n$ from the argument list of $K_n$ which can be reconstructed from $\{\Omega_1, ..., \Omega_{n-1}\}$.

Finally, we express $G_{A_1...A_n}(\omega_1, ..., \omega_n)$ of Eq. (11) using the kernel $\mathcal{K}_n$ so that the general $\Omega_k \in \mathbb{C}$ get replaced by $\Omega_k^{\underline{a}\,\underline{b}}$ of Eq. (12). For these, $\Omega_{p(1)}^{\underline{12}} + \Omega_{p(2)}^{\underline{23}} + ... + \Omega_{p(n)}^{\underline{n1}} = i(\omega_1 + \omega_2 + ... + \omega_n)$,

since the $E_k$ cancel pairwise. The structure of Eq. (8) (which followed from time translational invariance) implies that the terms proportional to $R_n$ are guaranteed to cancel when summed over permutations $p \in S_n$, so that only the terms proportional to $K_n$ remain. We drop the $\beta \delta_{0,\omega_1+\omega_2+...+\omega_n}$ from both sides [c.f. Eq. (8)] and find the spectral representation of the $n$-point correlation function in the Matsubara formalism,

$$G_{A_1...A_n}(\omega_1,...,\omega_{n-1}) = \frac{1}{Z} \sum_{p \in S_n} \zeta(p) \sum_{\underline{1...n}} e^{-\beta E_1} A_{p(1)}^{\underline{12}} A_{p(2)}^{\underline{23}} ... A_{p(n)}^{\underline{n1}} \times K_n \left( \Omega_{p(1)}^{\underline{12}}, \Omega_{p(2)}^{\underline{23}}, ..., \Omega_{p(n-1)}^{\underline{n-1\,n}} \right). \quad (15)$$

An equivalent expression was derived in the literature before [21], see also Refs. [18–20] for the cases of certain small $n$. However, kernel functions $K_n$ where previously only known approximately, for situations involving only a low order of anomalous terms, see the discussion in Sec. 5. We define an *anomalous* term of order $a = 1, 2, ... n-1$ as a summand contributing to $K_n(\Omega_1, ..., \Omega_{n-1})$ that contains a product of $a$ Kronecker delta functions $\delta_{0,x}$, where $x$ is a sum of a subset of $\{\Omega_1, ..., \Omega_{n-1}\}$. As can be seen in Fig. 1(c), these anomalous contributions to $G_{A_1...A_n}(\omega_1, ..., \omega_{n-1})$ correspond to qualitatively important sharp features.

In the next section, we present a simple, exact expression for general $K_n(\Omega_1, ..., \Omega_{n-1})$. Readers not interested in the derivation can directly skip to the result in Eq. (26) or its explicit form for $n = 2, 3, 4, 5$ in Sec. 5.

# 4 General kernel function $K_n(\Omega_1, ..., \Omega_{n-1})$

Assuming the spectrum and matrix elements entering Eq. (15) are known, the remaining task is to find expressions for the kernel function $K_n(\Omega_1, ..., \Omega_{n-1})$ defined via Eqns. (13) and (14) as the part of $\mathcal{K}_n(\Omega_1, \Omega_2, ..., \Omega_n)$ multiplying $\beta \delta_{0,\Omega_1+\Omega_2+...+\Omega_n}$. To facilitate the presentation in this section, in Eq. (13) we rename the integration variables $\tau_k \to \tau_{n-k+1}$ and define new arguments $z_{n-j+1} = \Omega_j$ for $j = 1, 2, ..., n-1$,

$$\mathcal{K}_n(\Omega_1 = z_n, \Omega_2 = z_{n-1}, ..., \Omega_n = z_1)$$
$$= \left[ \int_0^\beta d\tau_n e^{z_n \tau_n} \right] \left[ \int_0^{\tau_n} d\tau_{n-1} e^{z_{n-1}\tau_{n-1}} \right] ... \left[ \int_0^{\tau_3} d\tau_2 e^{z_2 \tau_2} \right] \left[ \int_0^{\tau_2} d\tau_1 \underbrace{e^{z_1 \tau_1}}_{\equiv h_1(\tau_1)} \right] \quad (16)$$
$$\underbrace{\phantom{\left[ \int_0^{\tau_2} d\tau_1 e^{z_1 \tau_1} \right]}}_{\equiv h_2(\tau_2)}$$

$$= \beta \delta_{0,z_1+z_2+...+z_n} K_n(z_n, z_{n-1}, ..., z_2) + R_n(z_n, z_{n-1}, ..., z_1). \quad (17)$$

As indicated in Eq. (16), we call $h_k(\tau_k)$ the integrand for the $\int_0^{\tau_{k+1}} d\tau_k$ integral for $k = 1, 2, ..., n$. At $k = 1$ this integrand is given by $h_1(\tau_1) = e^{z_1 \tau_1}$ and we will find $h_k$ for $k = 2, 3, ..., n$ iteratively. For $z \in \mathbb{C}$, we define the abbreviations $\delta_z \equiv \delta_{0,z}$ and

$$\Delta_z \equiv \begin{cases} 0, & \text{if } z = 0, \\ \frac{1}{z}, & \text{if } z \neq 0, \end{cases} \quad (18)$$

and consider the integral (for $p = 0, 1, 2, ...$ and $\tilde{\tau} \geq 0$, proof by partial integration and induction)

$$\int_0^{\tilde{\tau}} d\tau \, \tau^p e^{z\tau} = \left[ \frac{\tilde{\tau}^{p+1}}{p+1} \delta_z + p! (-1)^p \Delta_z^{1+p} \sum_{l=0}^p \frac{(-1)^l}{l!} \Delta_z^{-l} \tilde{\tau}^l \right] e^{z\tilde{\tau}} - p! (-1)^p \Delta_z^{p+1}. \quad (19)$$

Recall that we are only interested in the contribution $K_n(z_n, z_{n-1}, ..., z_2)$ that fulfills frequency conservation, see Eq. (17). The $\delta_{z_1+z_2+...+z_n}$ in front of this term arises from the final $\tau_n$ integration of $h_n(\tau_n) \propto e^{(z_1+z_2+...+z_n)\tau_n}$ via the first term in Eq. (19). This however requires

that all $z_k$ (except the vanishing ones, of course) remain in the exponent during the iterative integrations. This requirement is violated by the last term in the general integral (19) (which comes from the lower boundary of the integral). All terms in $\mathcal{K}_n$ that stem from this last term in Eq. (19) thus contribute to $R_n$ and can be dropped in the following [21]. Note however, that it is straightforward to generalize our approach and keep these terms if the full $\mathcal{K}_n$ is required.

To define the iterative procedure to solve the $n$-fold integral in Eq. (16), we make the ansatz

$$h_k(\tau_k) = \sum_{l=0}^{k-1} f_k(l)\tau_k^l e^{(z_k+z_{k-1}+...+z_1)\tau_k}, \tag{20}$$

which follows from the form of the integral (19) and our decision to disregard the terms contributing to $R_n$. The ansatz (20) is parameterized by the numbers $f_k(l)$ with $l = 0, 1, ..., k-1$. These numbers have to be determined iteratively, starting from $f_{k=1}(l=0) = 1$, read off from $h_1(\tau_1) = e^{z_1\tau_1}$, c.f. Eq. (16). Iteration rules to obtain the $f_k(l)$ from $f_{k-1}(l)$ are easily derived from Eqns. (16), (19) and (20). We obtain the recursion relation

$$f_k(l) = \sum_{p=0}^{k-1} \tilde{M}_{k-1}(l,p) f_{k-1}(p). \tag{21}$$

This can be understood as a matrix-vector product of $\mathbf{f}_{k-1} = (f_{k-1}(0), f_{k-1}(1), ..., f_{k-1}(k-2))^{\mathrm{T}}$ with the $k \times (k-1)$-matrix

$$\tilde{M}_{k-1}(l,p) = \frac{p!}{l!}\left[\delta_{l,p+1}\tilde{\delta}_{k-1} + \theta(p-l+1/2)(-1)^{l+p}\tilde{\Delta}_{k-1}^{1+p-l}\right], \tag{22}$$

where $\tilde{\Delta}_k \equiv \Delta_{z_k+...+z_2+z_1}$, $\tilde{\delta}_k \equiv \delta_{z_k+...+z_2+z_1}$. The tilde on top of the $\tilde{\delta}_k$ and $\tilde{\Delta}_k$ signals the presence of a sum of $z_j$ in the arguments (below we will define related quantities without tilde for the sum of $\Omega_j$). Note that the first (second) term in brackets of Eq. (22) comes from the first (second) term in square brackets of Eq. (19).

The next step is to find $K_n(z_n, z_{n-1}, ..., z_2)$. This requires to do the integral $\int_0^\beta d\tau_n h_n(\tau_n)$ which can be again expressed via Eq. (19) but with the replacement $\tilde{\tau} \to \beta$. Only the first term provides a $\beta\delta_{z_1+z_2+...+z_n}$ and is thus identified with $K_n$. We find:

$$K_n(z_n, z_{n-1}, ..., z_2) = \sum_{l=0}^{n-1} \frac{\beta^l f_n(l)}{l+1}. \tag{23}$$

The argument $z_1$ that the right hand side of Eq. (23) depends on is to be replaced by $z_1 = -z_2 - z_3 - ... - z_n$, in line with the arguments in $K_n(z_n, z_{n-1}, ..., z_2)$. Then, to conform with Eq. (15), we reinstate $\Omega_j = z_{n-j+1}$ for $j = 1, 2, ..., n-1$. This amounts to replacing the terms $\tilde{\delta}_j$ and $\tilde{\Delta}_j$ that appear in $f_n(l)$ as follows,

$$\tilde{\delta}_j = \delta_{z_j+...+z_2+z_1} = \delta_{\Omega_1+\Omega_2+...+\Omega_{n-j}} \equiv \delta_{n-j}, \tag{24}$$

$$-\tilde{\Delta}_j = -\Delta_{z_j+...+z_2+z_1} = \Delta_{\Omega_1+\Omega_2+...+\Omega_{n-j}} \equiv \Delta_{n-j}, \tag{25}$$

where we used $\Omega_1 + \Omega_2 + ... + \Omega_n = 0 = z_n + ... + z_2 + z_1$. Finally, we can express Eq. (23) using a product of $n-1$ matrices $\tilde{M}$ multiplying the initial length-1 vector with entry $f_1(0) = 1$. Transferring to the $\Omega$-notation by using Eqns. (24) and (25), we obtain

$$\boxed{\begin{aligned}&K_n(\Omega_1, ..., \Omega_{n-1})\\&= \sum_{i_{n-1}=0}^{n-1}\sum_{i_{n-2}=0}^{n-2}\cdots\sum_{i_2=0}^{2}\sum_{i_1=0}^{1}\frac{\beta^{i_{n-1}}}{i_{n-1}+1}M_1(i_{n-1},i_{n-2})M_2(i_{n-2},i_{n-3})\cdots M_{n-2}(i_2,i_1)M_{n-1}(i_1,0),\end{aligned}} \tag{26}$$

with

$$M_j(l,p) \equiv \frac{p!}{l!} \left[ \delta_{l,p+1} \delta_j - \theta(p-l+1/2) \Delta_j^{1+p-l} \right]. \tag{27}$$

The closed form expression (26) of the universal kernel, to be used in the spectral representation (15), is our main result. By definition it is free of any singularities as the case of vanishing denominators is explicitly excluded in Eq. (18).

## 5 Explicit kernel functions $K_n(\Omega_1, ..., \Omega_{n-1})$ for $n = 2, 3, 4, 5$

While the previous section gives a closed form expression for kernel functions of arbitrary order, we here evaluate the universal kernel functions $K_n(\Omega_1, ..., \Omega_{n-1})$ defined in Eq. (14) from Eq. (26) for $n = 2, 3, 4, 5$ and show the results in Tab. 1. In each column, the kernel function in the top row is obtained by first multiplying the entries listed below it in the same column by the common factor in the rightmost column and then taking the sum. The symbols $\delta_j$ and $\Delta_j$ for $j = 1, 2, ..., n-1$ which appear in Tab. 1 are defined by

$$\delta_j \equiv \delta_{\Omega_1 + \Omega_2 + ... + \Omega_j, 0}, \tag{28}$$

$$\Delta_j \equiv \Delta_{\Omega_1 + \Omega_2 + ... + \Omega_j} \equiv \begin{cases} 0, & \text{if } \Omega_1 + \Omega_2 + ... + \Omega_j = 0, \\ \frac{1}{\Omega_1 + \Omega_2 + ... + \Omega_j}, & \text{if } \Omega_1 + \Omega_2 + ... + \Omega_j \neq 0, \end{cases} \tag{29}$$

compare also to the previous section. As an example, for $n = 2$ and $n = 3$ we obtain from Tab. 1

$$K_2(\Omega_1) = -\Delta_{\Omega_1} + \frac{\beta}{2} \delta_{\Omega_1}, \tag{30}$$

$$K_3(\Omega_1, \Omega_2) = +\Delta_{\Omega_1} \Delta_{\Omega_1 + \Omega_2} - \frac{\beta}{2} \delta_{\Omega_1} \Delta_{\Omega_2} - \Delta_{\Omega_1} \delta_{\Omega_1 + \Omega_2} \left( \frac{\beta}{2} + \Delta_{\Omega_1} \right) + \delta_{\Omega_1} \delta_{\Omega_2} \frac{\beta}{2} \frac{\beta}{3}, \tag{31}$$

respectively. The rows of Tab. 1 are organized with respect to the number $a$ of factors $\delta_l$ in the summands. Here, $a = 0$ indicates the regular part and $a = 1, 2, ..., n-1$ indicates anomalous terms. There are $n-1$ *choose* $a$ anomalous terms of order $a$. Our results are exact and go substantially beyond existing expressions in the literature – these are limited to $n \leq 3$ [18] or to fermionic $n = 4$ [19–21] with $a = 0, 1$ (and $a = 2, 3$ guaranteed to vanish, see below) or arbitrary $n$ with $a = 0$ [21]. Alternative expressions for the $n = 3, 4$ kernel functions with $a \leq 1$ were given in [21], but they are consistent with our kernel functions as they yield the same correlation functions, see the Appendix.

In the case of purely fermionic correlators (all $A_k$ fermionic), individual Matsubara frequencies $\omega_k$ cannot be zero and thus the $\Omega_k^{\underline{a}\,\underline{b}} \equiv i\omega_k + E_{\underline{a}} - E_{\underline{b}}$ of Eq. (12) always have a finite imaginary part and are non-zero, regardless of the eigenenergies. In this case, only sums of an even number of frequencies can be zero, and we can simplify $\delta_1 = \delta_3 = \delta_5 = ... = 0$. The expressions for the kernels in Tab. 1, now denoted by $K_n|_F$ for the fermionic case, simplify to

$$K_2(\Omega_1)|_F = -\Delta_1, \tag{32}$$

$$K_4(\Omega_1, \Omega_2, \Omega_3)|_F = \Delta_1 \Delta_3 \left[ \delta_2 \left( \frac{\beta}{2} + \Delta_1 \right) - \Delta_2 \right], \tag{33}$$

$$K_6(\Omega_1, ..., \Omega_5)|_F = \Delta_1 \Delta_3 \Delta_5 \left\{ -\Delta_2 \Delta_4 - \delta_2 \delta_4 \left[ \frac{\beta}{2} \frac{\beta}{3} + (\Delta_1 + \Delta_3) \left( \frac{\beta}{2} + \Delta_1 \right) \right] \right. \tag{34}$$

$$\left. + \delta_4 \Delta_2 \left( \frac{\beta}{2} + \Delta_1 + \Delta_2 + \Delta_3 \right) + \delta_2 \Delta_4 \left( \frac{\beta}{2} + \Delta_1 \right) \right\}.$$

Table 1: Universal kernel functions $K_n\big(\Omega_1,...,\Omega_{n-1}\big)$ for $n = 2, 3, 4, 5$ defined in Eq. (14) and calculated from Eq. (26) in Sec. 4. In each column, the kernel function in the top row is obtained by first multiplying the entries listed below it in the same column by the common factor in the rightmost column and then taking the sum, see Eqns. (30) and (31) as examples. The symbols $\delta_j$ and $\Delta_j$ appearing are defined in Eqns. (28) and (29). The rows are organized with respect to the number $a$ of appearances of $\delta_j$, i.e. the order of the anomalous terms.

| #anom. | $K_2(\Omega_1)$ | $K_3(\Omega_1,\Omega_2)$ | $K_4(\Omega_1,\Omega_2,\Omega_3)$ | $K_5(\Omega_1,\Omega_2,\Omega_3,\Omega_4)$ | factor for entire row |
|---|---|---|---|---|---|
| $a=0$ | $-\Delta_1$ | $+\Delta_1\Delta_2$ | $-\Delta_1\Delta_2\Delta_3$ | $+\Delta_1\Delta_2\Delta_3\Delta_4$ | $1$ |
| | $+\delta_1$ | $-\delta_1\Delta_2$ | $+\delta_1\Delta_2\Delta_3$ | $-\delta_1\Delta_2\Delta_3\Delta_4$ | $\frac{\beta}{2}$ |
| | | $-\Delta_1\delta_2$ | $+\Delta_1\delta_2\Delta_3$ | $-\Delta_1\delta_2\Delta_3\Delta_4$ | $\frac{\beta}{2}+\Delta_1$ |
| $a=1$ | | | $+\Delta_1\Delta_2\delta_3$ | $-\Delta_1\Delta_2\delta_3\Delta_4$ | $\frac{\beta}{2}+\Delta_1+\Delta_2$ |
| | | | | $-\Delta_1\Delta_2\Delta_3\delta_4$ | $\frac{\beta}{2}+\Delta_1+\Delta_2+\Delta_3$ |
| | | $+\delta_1\delta_2$ | $-\delta_1\delta_2\Delta_3$ | $+\delta_1\delta_2\Delta_3\Delta_4$ | $\frac{\beta}{2}\frac{\beta}{3}$ |
| | | | $-\delta_1\Delta_2\delta_3$ | $+\delta_1\Delta_2\delta_3\Delta_4$ | $\frac{\beta}{2}\left(\frac{\beta}{3}+\Delta_2\right)$ |
| $a=2$ | | | $-\Delta_1\delta_2\delta_3$ | $+\Delta_1\delta_2\delta_3\Delta_4$ | $\frac{\beta}{2}\frac{\beta}{3}+\Delta_1\left(\frac{\beta}{2}+\Delta_1\right)$ |
| | | | | $+\delta_1\Delta_2\Delta_3\delta_4$ | $\frac{\beta}{2}\left(\frac{\beta}{3}+\Delta_2+\Delta_3\right)$ |
| | | | | $+\Delta_1\delta_2\Delta_3\delta_4$ | $\frac{\beta}{2}\frac{\beta}{3}+(\Delta_1+\Delta_3)\left(\frac{\beta}{2}+\Delta_1\right)$ |
| | | | | $+\Delta_1\Delta_2\delta_3\delta_4$ | $\frac{\beta}{2}\frac{\beta}{3}+(\Delta_1+\Delta_2)\left(\frac{\beta}{2}+\Delta_2\right)+\Delta_1^2$ |
| | | | $+\delta_1\delta_2\delta_3$ | $-\delta_1\delta_2\delta_3\Delta_4$ | $\frac{\beta}{2}\frac{\beta}{3}\frac{\beta}{4}$ |
| $a=3$ | | | | $-\delta_1\delta_2\Delta_3\delta_4$ | $\frac{\beta}{2}\frac{\beta}{3}\left(\frac{\beta}{4}+\Delta_3\right)$ |
| | | | | $-\delta_1\Delta_2\delta_3\delta_4$ | $\frac{\beta}{2}\left(\frac{\beta}{3}\frac{\beta}{4}+\Delta_2\left(\frac{\beta}{3}+\Delta_2\right)\right)$ |
| | | | | $-\Delta_1\delta_2\delta_3\delta_4$ | $\frac{\beta}{2}\frac{\beta}{3}\frac{\beta}{4}+\Delta_1\left(\frac{\beta}{2}\frac{\beta}{3}+\Delta_1\left(\frac{\beta}{2}+\Delta_1\right)\right)$ |
| $a=4$ | | | | $+\delta_1\delta_2\delta_3\delta_4$ | $\frac{\beta}{2}\frac{\beta}{3}\frac{\beta}{4}\frac{\beta}{5}$ |

This concludes the general part of this work. Next, we consider two example systems frequently discussed in the condensed matter theory literature. Using our formalism, we provide analytical forms of correlation functions that to the best of our knowledge were not available before.

# 6 Applications: Hubbard atom and free spin

## 6.1 Fermionic Hubbard atom

The Hubbard atom (HA) describes an isolated impurity or otherwise localized system with Hamiltonian

$$H = \epsilon\left(n_\uparrow + n_\downarrow\right) + U n_\uparrow n_\downarrow - h\left(n_\uparrow - n_\downarrow\right), \tag{35}$$

see Fig. 1(b) for a sketch. The HA corresponds to the limit of vanishing system-bath coupling of the Anderson impurity model (AIM), or vanishing hopping in the Hubbard model (HM). The particle number operators $n_\sigma = d_\sigma^\dagger d_\sigma$ count the number of fermionic particles with spin $\sigma \in \{\uparrow, \downarrow\}$, each contributing an onsite energy $\epsilon$ shifted by an external magnetic field $h$ in $z$-direction. An interaction energy $U$ is associated to double occupation.

Due to its simplicity and the four-dimensional Hilbert space, the correlation functions for the HA can be found analytically using the spectral representation. It is therefore often used for benchmarking [3, 27, 28]. The presence of the interaction term leads to a non-vanishing $n = 4$ one-line irreducible vertex function. The HA serves as an important reference point to study and interpret properties of the AIM and HM beyond the one-particle level, for example diver-

gences of two-line irreducible vertex functions [29–32] and signatures of the local moment formation in generalized susceptibilities [16, 33]. Using the fermionic kernels in Eqns. (32) and (33), we have checked that our formalism reproduces the results for the 2-point and 4-point correlators given in Refs. [19, 21, 26] for half-filling, $\epsilon = -U/2$ and $h = 0$.

Correlation functions including bosonic operators describe the asymptotic behaviour of the $n = 4$ fermion vertex for large frequencies [34] or the interaction of electrons by the exchange of an effective boson [35, 36]. These relations involve correlation functions of two bosonic operators or of one bosonic and two fermionic operators, giving rise to expressions possibly anomalous in at most one frequency argument, i.e. $a \leq 1$.

For the HA, AIM and HM, bosonic correlation functions for $n > 2$ have not been considered thoroughly so far. Only recently, steps in this direction were taken, particularly in the context of non-linear response theory [3]. The response of a system to first and second order in an external perturbation is described by 2- and 3-point correlation functions, respectively. For the HA, physically motivated perturbations affect the onsite energy via a term $\delta_\epsilon n$ or take the form of a magnetic field $\boldsymbol{\delta}_h \cdot \mathbf{S}$. Here, the parameters $\delta_\epsilon$ and $\boldsymbol{\delta}_h$ denote the strength of the perturbation and we define

$$n = n_\uparrow + n_\downarrow, \qquad S^x = \frac{1}{2}\left(d_\uparrow^\dagger d_\downarrow + d_\downarrow^\dagger d_\uparrow\right), \quad S^y = \frac{-i}{2}\left(d_\uparrow^\dagger d_\downarrow - d_\downarrow^\dagger d_\uparrow\right), \quad S^z = \frac{1}{2}\left(n_\uparrow - n_\downarrow\right). \tag{36}$$

The resulting changes of the expectation values of the density or magnetization in arbitrary direction are described in second order of the perturbation by the connected parts of the correlation functions $G_{A_1 A_2 A_3}(\tau_1, \tau_2, \tau_3)$, with $A_i \in \{n, S_x, S_y, S_z\}$, where the time-ordered expectation value is evaluated with respect to the unperturbed system (35) and Fourier transformed to the frequencies of interest. These objects have been studied numerically in Ref. [3]. In the following, we give explicit, analytic expressions of the full correlation functions $G_{A_1 A_2 A_3}(\omega_1, \omega_2)$ (i.e. including disconnected parts), for arbitrary parameters $\epsilon$, $U$ and $h$ and for all possible operator combinations using the (bosonic) kernel function $K_3$, see Eq. (31). To the best of our knowledge, these expressions have not been reported before.

The eigenstates of the HA Hamiltonian (35) [see Fig. 1(b)] describe an empty ($|0\rangle$), singly occupied ($d_\uparrow^\dagger|0\rangle = |\uparrow\rangle$, $d_\downarrow^\dagger|0\rangle = |\downarrow\rangle$) or doubly occupied ($d_\uparrow^\dagger d_\downarrow^\dagger|0\rangle = |\uparrow\downarrow\rangle$) impurity with eigenenergies $E_0 = 0$, $E_\uparrow = \epsilon - h$, $E_\downarrow = \epsilon + h$ and $E_{\uparrow\downarrow} = 2\epsilon + U$, respectively. The partition function is $Z = 1 + e^{-\beta(\epsilon-h)} + e^{-\beta(\epsilon+h)} + e^{-\beta(2\epsilon+U)}$. We define

$$s = \frac{e^{-\beta\epsilon}}{Z}\sinh(\beta h), \qquad c = \frac{e^{-\beta\epsilon}}{Z}\cosh(\beta h), \tag{37}$$

and obtain all non-vanishing bosonic 3-point correlation functions (where $\omega_3 = -\omega_1 - \omega_2$):

$$G_{nnn}(\omega_1, \omega_2) = 2\beta^2 \delta_{\omega_1}\delta_{\omega_2}\left(\frac{4e^{-\beta(2\epsilon+U)}}{Z} + c\right), \tag{38}$$

$$G_{nnS^z}(\omega_1, \omega_2) = \beta^2 \delta_{\omega_1}\delta_{\omega_2} s, \tag{39}$$

$$G_{nS^x S^y}(\omega_1, \omega_2) = -\beta \delta_{\omega_1} s \frac{\omega_2}{\omega_2^2 + 4h^2}, \tag{40}$$

$$G_{nS^x S^x}(\omega_1, \omega_2) = G_{nS^y S^y}(\omega_1, \omega_2) = 2\beta \delta_{\omega_1}\frac{h\,s}{\omega_2^2 + 4h^2}, \tag{41}$$

$$G_{nS^z S^z}(\omega_1, \omega_2) = \frac{\beta^2}{2}\delta_{\omega_1}\delta_{\omega_2} c, \tag{42}$$

$$G_{S^z S^x S^x}(\omega_1, \omega_2) = G_{S^z S^y S^y}(\omega_1, \omega_2) = -s\frac{\omega_2\omega_3 + 4h^2}{(\omega_2^2 + 4h^2)(\omega_3^2 + 4h^2)} + \beta\delta_{\omega_1}\frac{h\,c}{\omega_2^2 + 4h^2}, \tag{43}$$

$$G_{S^z S^z S^z}(\omega_1, \omega_2) = \frac{\beta^2}{4} \delta_{\omega_1} \delta_{\omega_2} s \, , \tag{44}$$

$$G_{S^x S^y S^z}(\omega_1, \omega_2) = 2h \, s \, \frac{\omega_1 - \omega_2}{(\omega_1^2 + 4h^2)(\omega_2^2 + 4h^2)} - \frac{\beta}{2} \delta_{\omega_3} c \, \frac{\omega_1}{\omega_1^2 + 4h^2} \, . \tag{45}$$

We observe that each conserved quantity, in this case $n$ and $S_z$, contributes an anomalous term $\propto \delta_{\omega_k}$ in its respective frequency argument $\omega_k$. If an operator $A_k$ is conserved $[H, A_k] = 0$, the basis over which we sum in Eq. (15) can be chosen such that both $H$ and $A_k$ are diagonal, $A_k^{12} = A_k^{11} \delta_{1,2}$. If $A_k^{11} \neq 0$ for some state $|\underline{1}\rangle$ the vanishing eigenenergy difference leads to the appearance of an anomalous contribution. If the operators in the correlator additionally commute with each other, in our case for example $[n, S^z] = 0$, there exists a basis in which all operators and the Hamiltonian are diagonal, giving rise to correlation functions anomalous in all frequency arguments.

In the limit of vanishing field $h \to 0$, we introduce an additional degeneracy $E_\uparrow = E_\downarrow = \epsilon$ in the system, potentially resulting in additional anomalous contributions. The corresponding correlation functions can then be obtained in two ways. Either we recompute them using the kernel function $K_3$ or we take appropriate limits, for example

$$\lim_{h \to 0} \frac{h \, \sinh(\beta h)}{\omega_k^2 + 4h^2} = \frac{\beta}{4} \delta_{\omega_k} \, , \tag{46}$$

resulting in

$$G_{nnn}(\omega_1, \omega_2) = \beta^2 \delta_{\omega_1} \delta_{\omega_2} \frac{2 \left( 4 e^{-\beta(2\epsilon + U)} + e^{-\beta\epsilon} \right)}{Z} \, , \tag{47}$$

$$G_{n S^\alpha S^\alpha}(\omega_1, \omega_2) = \beta^2 \delta_{\omega_1} \delta_{\omega_2} \frac{e^{-\beta\epsilon}}{2Z} \quad (\alpha \in \{x, y, z\}) \, , \tag{48}$$

$$G_{S^x S^y S^z}(\omega_1, \omega_2) = \beta \frac{e^{-\beta\epsilon}}{2Z} \left( -\delta_{\omega_1} \Delta_{\omega_2} + \delta_{\omega_2} \Delta_{\omega_1} - \delta_{\omega_1 + \omega_2} \Delta_{\omega_1} \right) \, , \tag{49}$$

with all other correlation functions vanishing. As already pointed out in Ref. [3], only the last correlation function retains a nontrivial frequency dependence due to non-commuting operators.

## 6.2 Free spin $S$

We now consider correlation functions of a free spin of length $S$, without a magnetic field, so that temperature $T = 1/\beta$ is the only energy scale. The operators $\{S^\alpha\}_{\alpha = x,y,z}$ fulfill $S^x S^x + S^y S^y + S^z S^z = S(S+1)$ and the SU(2) algebra $[S^{\alpha_1}, S^{\alpha_2}] = i \sum_{\alpha_3 = \{x,y,z\}} \epsilon^{\alpha_1 \alpha_2 \alpha_3} S^{\alpha_3}$, thus they are bosonic. Since the Hamiltonian vanishes and therefore all eigenenergies are zero, every $\Omega_k^{\frac{a\,b}{}}$ in the spectral representation (15) can vanish and a proper treatment of all anomalous terms is essential. As the Heisenberg time dependence is trivial, $S^\alpha(\tau) = S^\alpha$, the non-trivial frequency dependence of the correlators, which can be non-vanishing at any order $n > 1$, derives solely from the action of imaginary time-ordering.

The correlators are required, for example, as the non-trivial initial condition for the spin-fRG recently suggested by Kopietz et al., Refs. [13, 37–40]. However, for $n > 3$ they are so far only partially available: They are either given for restricted frequency combinations, or for the purely classical case $S^{\alpha_1} = S^{\alpha_2} = \ldots = S^{\alpha_n}$ where the SU(2) algebra does not matter, or for finite magnetic field via an equation of motion [37] or diagrammatic approach [41, 42].

Table 2: Matsubara correlation functions for a free spin-$S$ up to order $n = 4$. Here, $\omega_4 = -\omega_1 - \omega_2 - \omega_3$.

| | |
|---|---|
| $n = 2$ | $G_{S^+ S^-}(\omega) = G_{S^z S^z}(\omega) = \beta \delta_\omega b_1$ |
| $n = 3$ | $G_{S^+ S^- S^z}(\omega_1, \omega_2) = \beta b_1 (-\delta_{\omega_1} \Delta_{i\omega_2} + \delta_{\omega_2} \Delta_{i\omega_1} + \delta_{\omega_1 + \omega_2} \Delta_{i\omega_2}) = -i G_{S^x S^y S^z}(\omega_1, \omega_2)$ |
| $n = 4$ | $G_{S^z S^z S^z S^z}(\omega_1, \omega_2, \omega_3) = \delta_{\omega_1} \delta_{\omega_2} \delta_{\omega_3} \beta^3 b_3$ |
| | $G_{S^+ S^+ S^- S^-}(\omega_1, \omega_2, \omega_3) = \beta b_1 [2 \times \delta_{\omega_1} \delta_{\omega_2} \delta_{\omega_3} \times \frac{\beta^2}{5} (3 b_1 - \frac{1}{3}) + r]$ |
| | $G_{S^+ S^- S^z S^z}(\omega_1, \omega_2, \omega_3) = \beta b_1 [1 \times \delta_{\omega_1} \delta_{\omega_2} \delta_{\omega_3} \times \frac{\beta^2}{5} (3 b_1 - \frac{1}{3}) - r]$ |
| | $r = \Delta_{i\omega_1} \Delta_{i\omega_2} (\delta_{\omega_1 + \omega_3} + \delta_{\omega_2 + \omega_3} - \delta_{\omega_3} - \delta_{\omega_4}) - (\delta_{\omega_1} \Delta_{i\omega_2}^2 + \delta_{\omega_2} \Delta_{i\omega_1}^2)(\delta_{\omega_3} + \delta_{\omega_4})$ |
| | $\quad - \Delta_{i\omega_3} \Delta_{i\omega_4} (\delta_{\omega_1} + \delta_{\omega_2})$ |

We define the spin raising and lowering operators,

$$S^{\pm} = (S^x \pm i S^y)/\sqrt{2}, \tag{50}$$

which have to appear in pairs for a non-vanishing correlator due to spin-rotation symmetry. As for the HA, we do not consider connected correlators in this work for brevity. The classical $S^z$-correlator can be found from its generating functional with source field $h$ [13],

$$\mathcal{G}(y = \beta h) = \frac{\sinh[y(S + 1/2)]}{(2S + 1)\sinh[y/2]}, \tag{51}$$

$$\langle (S^z)^l \rangle = \lim_{y \to 0} \partial_y^l \mathcal{G}(y) \equiv b_{l-1}, \tag{52}$$

for example $b_1 = \frac{S}{3}(S + 1)$ and $b_3 = \frac{S}{15}(3 S^3 + 6 S^2 + 2 S - 1)$ and vanishing $b_l$ for even $l$. For all other correlators involving $\alpha_k = \pm$, we adapt Eq. (15) for the free spin case,

$$G_{S^{\alpha_1} S^{\alpha_2} \dots S^{\alpha_n}}(\omega_1, \dots, \omega_{n-1}) = \sum_{p \in S_n} \langle S^{\alpha_{p(1)}} S^{\alpha_{p(2)}} \dots S^{\alpha_{p(n)}} \rangle K_n(i\omega_{p(1)}, i\omega_{p(2)}, \dots, i\omega_{p(n-1)}), \tag{53}$$

where we made use of the fact that all eigenenergies are zero and the Heisenberg time evolution is trivial. It is convenient to evaluate the equal-time correlators in Eq. (53) as

$$\langle S^{\alpha_1} S^{\alpha_2} \dots S^{\alpha_n} \rangle = \frac{1}{2S + 1} \sum_{m=-S}^{S} \langle m | S^{\alpha_1} S^{\alpha_2} \dots S^{\alpha_n} | m \rangle \equiv \frac{1}{2S + 1} \sum_{m=-S}^{S} \sum_{l=0}^{n} p_l m^l = p_0 + \sum_{l=2}^{n} p_l b_{l-1}, \tag{54}$$

where in the last step we used Eq. (52). We find the real expansion coefficients $\{p_l\}_{l=0,1,\dots,n}$ iteratively by moving through the string $\alpha_1 \alpha_2 \dots \alpha_n$ from the right and start from $p_l = \delta_{0,l}$. Based on the $S^z$ eigenstates $\{|m\rangle\}_{m=-S,\dots,S-1,S}$ we obtain the iteration rules from $S^z |m\rangle = m |m\rangle$ and $S^{\pm} |m\rangle = \sqrt{1/2} \sqrt{S(S + 1) - m(m \pm 1)} |m \pm 1\rangle$. We define an auxiliary integer $c$ that keeps track of the intermediate state $|m + c\rangle$, initially $c = 0$. Depending on the $\alpha_j$ that we find in step $j = n, n-1 \dots, 1$ we take one of the following actions: (i) For $\alpha_j = z$, we update $p_l \leftarrow p_{l-1} + c p_l \; \forall l$ and leave $c$ unchanged. It is understood that $p_{l<0} = 0$. (ii) For $\alpha_j = +$, we combine the square-root factor brought by the raising operator with the factor that comes from the necessary $\alpha_{j'} = -$ at another place in the string. We replace $p_l \leftarrow -\frac{1}{2} p_{l-2} - \frac{2c+1}{2} p_{l-1} + (\frac{3}{2} b_1 - c \frac{c+1}{2}) p_l \; \forall l$ and then let $c \leftarrow c + 1$. (iii) For $\alpha_j = -$, we update $c \leftarrow c - 1$ and keep $p_l$ unchanged, $p_l \leftarrow p_l \; \forall l$.

Our final results for the free spin correlators are reported in Tab. 2. We reproduce the known spin correlators for $n = 2, 3$ and determine the non-classical correlators $G_{S^+ S^+ S^- S^-}$ and $G_{S^+ S^- S^z S^z}$ at order $n = 4$, which to the best of our knowledge were not available in the

literature.[1] We also confirmed the classical result for $G_{S^z S^z S^z S^z}$, which in our full quantum formalism requires some non-trivial cancellations. To arrive at our results, we used the identity

$$\Delta_{a+b}(\Delta_a + \Delta_b) - \Delta_a \Delta_b = \delta_a \Delta_b^2 + \delta_b \Delta_a^2 - \delta_{a+b} \Delta_a \Delta_b \,. \tag{55}$$

We finally comment on the relation between the $n = 3$ free spin-$S$ correlator $G_{S^+ S^- S^z}$ from Tab. (2) and the result for $G_{S^x S^y S^z}$ found for the zero-field limit of the HA in Eq. (49). The operators $S^{x,y,z}$ for the Hubbard model [c.f. Eq. (36)] project to the singly-occupied $S = 1/2$ subspace spanned by the states $\left|\uparrow\right\rangle, \left|\downarrow\right\rangle$. Thus, using $G_{S^x S^y S^z} = iG_{S^+ S^- S^z}$ and specializing the free spin result from Tab. (2) to $S = 1/2$ (where $b_1 = 1/4$) we find agreement with the HA result (49) up to the factor $2e^{-\beta\epsilon}/Z$. This factor represents the expectation value of the projector to the singly-occupied sector in the HA Hilbert space and goes to unity in the local-moment regime.

# 7 Conclusion

In summary, we have provided exact universal kernel functions for the spectral representation of the $n$-point Matsubara correlator. Our results are an efficient alternative to equation-of-motion approaches which often have difficulties to capture anomalous terms related to conserved or commuting operators. We expect our results to be useful for various benchmarking applications, as starting points for emerging many-body methods and for unraveling the physical interpretation of $n$-point functions in various settings. Our results also apply in the limit $T \to 0$ where the formally divergent anomalous contributions are to be understood as $\beta\delta_{\omega,0} \to 2\pi\delta(\omega)$. Some of these Dirac delta-functions will vanish after subtracting the disconnected contributions, others indicate truely divergent susceptibilities like the $1/T$ Curie law for the spin-susceptiblity of the Hubbard atom in the local moment regime [26]. Although our work has focused on imaginary frequency (Matsubara) correlators, with analytical expressions now at hand, it is also interesting to study the intricacies of analytical continuation to real frequencies and thus to further explore the connection of Matsubara and Keldysh correlators [43].

## Acknowledgements

We acknowledge useful discussions with Karsten Held, Friedrich Krien, Seung-Sup Lee, Peter Kopietz, Fabian Kugler, Nepomuk Ritz, Georg Rohringer, Andreas Rückriegel. We thank Andreas Rückriegel for sharing unpublished results on 4-point free spin correlators and pointing out further simplifications of the analytical expressions.

**Funding information** BS and BS are supported by a MCQST-START fellowship. We acknowledge funding from the International Max Planck Research School for Quantum Science and Technology (IMPRS-QST) for JH, from the Deutsche Forschungsgemeinschaft under Germany's Excellence Strategy EXC-2111 (Project No. 390814868), and from the Munich Quantum Valley, supported by the Bavarian state government with funds from the Hightech Agenda Bayern Plus.

---

[1]We thank Andreas Rückriegel for sharing unpublished results on 4-point free spin correlators and pointing out further simplifications of the analytical expressions.

## A  Equivalence to convention of Ref. [21]

In Ref. [21] by Kugler, Lee and von Delft (KLD), only regular ($a = 0$) and anomalous terms of order $a = 1$ have been considered for $n = 3, 4$. The corresponding kernel functions were derived from only $(n-1)!$ permutations by setting $\tau_n = 0$ and $\tau_{i \neq n} > 0$, but still applied to all $n!$ permutations to obtain the correlation functions. For $n = 3$, the resulting kernel function (Eq. (46) in Ref. [21]) reads

$$K_{3,\text{KLD}}(\Omega_1, \Omega_2) = \Delta_1 \Delta_2 - \Delta_1 \delta_2 \frac{1}{2}(\beta + \Delta_1) - \delta_1 \Delta_2 \frac{1}{2}(\beta + \Delta_2). \tag{A.1}$$

This can be compared to the corresponding kernel function for $n = 3$ found in our Eq. (31) truncated to $a \leq 1$,

$$K_3^{a \leq 1}(\Omega_1, \Omega_2) = \Delta_1 \Delta_2 - \Delta_1 \delta_2 \left( \frac{\beta}{2} + \Delta_1 \right) - \frac{\beta}{2} \delta_1 \Delta_2. \tag{A.2}$$

Both approaches are equally valid and should yield the same correlation functions (consistently discarding terms with $a = 2$), yet the kernel functions are obviously different. To resolve this issue, we define the difference of the kernel functions

$$K_{3,\text{diff}}(\Omega_1, \Omega_2) = K_{3,\text{KLD}}(\Omega_1, \Omega_2) - K_3^{a \leq 1}(\Omega_1, \Omega_2) = \frac{1}{2} \left( \Delta_1^2 \delta_2 - \delta_1 \Delta_2^2 \right), \tag{A.3}$$

and show that the corresponding contributions to the correlation function vanishes when summed over cyclically related permutations $p = 123, 231, 312$. These contributions are given by

$$
\begin{aligned}
&\frac{1}{Z} \sum_{p=123,231,312} \zeta(p) \sum_{\underline{123}} e^{-\beta E_{\underline{1}}} A_{p(1)}^{1\underline{2}} A_{p(2)}^{2\underline{3}} A_{p(3)}^{3\underline{1}} K_{3,\text{diff}}(\Omega_{p(1)}^{1\underline{2}}, \Omega_{p(2)}^{2\underline{3}}) \\
&= \frac{\zeta(123)}{2Z} \sum_{\underline{123}} e^{-\beta E_{\underline{1}}} A_{1}^{1\underline{2}} A_{2}^{2\underline{3}} A_{3}^{3\underline{1}} \left( \Delta_{\Omega_{\underline{1}}^{1\underline{2}}}^2 \delta_{\Omega_{\underline{1}}^{1\underline{2}} + \Omega_{\underline{2}}^{2\underline{3}}} - \delta_{\Omega_{\underline{1}}^{1\underline{2}}} \Delta_{\Omega_{\underline{1}}^{1\underline{2}} + \Omega_{\underline{2}}^{2\underline{3}}}^2 \right) \\
&+ \frac{\zeta(231)}{2Z} \sum_{\underline{123}} e^{-\beta E_{\underline{1}}} A_{2}^{1\underline{2}} A_{3}^{2\underline{3}} A_{1}^{3\underline{1}} \left( \Delta_{\Omega_{\underline{2}}^{1\underline{2}}}^2 \delta_{\Omega_{\underline{2}}^{1\underline{2}} + \Omega_{\underline{3}}^{2\underline{3}}} - \delta_{\Omega_{\underline{2}}^{1\underline{2}}} \Delta_{\Omega_{\underline{2}}^{1\underline{2}} + \Omega_{\underline{3}}^{2\underline{3}}}^2 \right) \\
&+ \frac{\zeta(312)}{2Z} \sum_{\underline{123}} e^{-\beta E_{\underline{1}}} A_{3}^{1\underline{2}} A_{1}^{2\underline{3}} A_{2}^{3\underline{1}} \left( \Delta_{\Omega_{\underline{3}}^{1\underline{2}}}^2 \delta_{\Omega_{\underline{3}}^{1\underline{2}} + \Omega_{\underline{1}}^{2\underline{3}}} - \delta_{\Omega_{\underline{3}}^{1\underline{2}}} \Delta_{\Omega_{\underline{3}}^{1\underline{2}} + \Omega_{\underline{1}}^{2\underline{3}}}^2 \right).
\end{aligned}
\tag{A.4}
$$

Considering the second term of permutation $p = 312$ and renaming the summation variables $\underline{2} \to \underline{1}, \underline{3} \to \underline{2}, \underline{1} \to \underline{3}$ yields

$$
\begin{aligned}
&- \frac{\zeta(312)}{2Z} \sum_{\underline{123}} e^{-\beta E_{\underline{1}}} A_{3}^{1\underline{2}} A_{1}^{2\underline{3}} A_{2}^{3\underline{1}} \delta_{\Omega_{\underline{3}}^{1\underline{2}}} \Delta_{\Omega_{\underline{3}}^{1\underline{2}} + \Omega_{\underline{1}}^{2\underline{3}}}^2 \\
&= - \frac{\zeta(312)}{2Z} \sum_{\underline{123}} e^{-\beta E_{\underline{3}}} A_{1}^{1\underline{2}} A_{2}^{2\underline{3}} A_{3}^{3\underline{1}} \delta_{\Omega_{\underline{3}}^{3\underline{1}}} \Delta_{\Omega_{\underline{3}}^{3\underline{1}} + \Omega_{\underline{1}}^{1\underline{2}}}^2 \\
&= - \frac{\zeta(123)}{2Z} \sum_{\underline{123}} e^{-\beta E_{\underline{1}}} A_{1}^{1\underline{2}} A_{2}^{2\underline{3}} A_{3}^{3\underline{1}} \delta_{\Omega_{\underline{1}}^{1\underline{2}} + \Omega_{\underline{2}}^{2\underline{3}}} \Delta_{\Omega_{\underline{1}}^{1\underline{2}}}^2,
\end{aligned}
\tag{A.5}
$$

where we used $\omega_3 = -\omega_1 - \omega_2$ and the fact that $\delta_{\Omega_{\underline{3}}^{3\underline{1}}} = \delta_{\omega_3} \delta_{E_{\underline{1}}, E_{\underline{3}}}$ enforces the third operator to be bosonic, such that $\zeta(312) = \zeta(123)$. This term exactly cancels the first contribution of permutation $p = 123$ in (A.4). Repeating similar steps for the remaining terms, we find that

the the second term of $p = 123$ and the first term of $p = 231$ as well as the second term of $p = 231$ and the first term of $p = 312$ cancel, leading to

$$\frac{1}{Z} \sum_{p \in \{123,231,312\}} \zeta(p) \sum_{\underline{123}} e^{-\beta E_1} A_{p(1)}^{\underline{12}} A_{p(2)}^{\underline{23}} A_{p(3)}^{\underline{31}} K_{3,\text{diff}}\left(\Omega_{p(1)}^{\underline{12}}, \Omega_{p(2)}^{\underline{23}}\right) = 0 \,. \tag{A.6}$$

Similarly, summing over the second set of cyclically related permutations $p = 132, 213, 321$ leads to a vanishing result, leading to the conclusion that

$$\frac{1}{Z} \sum_{p \in S_3} \zeta(p) \sum_{\underline{123}} e^{-\beta E_1} A_{p(1)}^{\underline{12}} A_{p(2)}^{\underline{23}} A_{p(3)}^{\underline{31}} K_{3,\text{diff}}\left(\Omega_{p(1)}^{\underline{12}}, \Omega_{p(2)}^{\underline{23}}\right) = 0 \,. \tag{A.7}$$

Thus we have shown that both kernel functions in Eqns. (A.1) and (A.2) are equivalent as they yield the same correlation functions after summing over all permutations. The same statement holds true for case of $n = 4$ and $a = 1$.

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
