# Peer review of "Spectral representation of Matsubara n-point functions: Exact kernel functions and applications"

_SciPost Physics, doi:SciPost Phys. 15, 183 (2023)_

## Round 1 · Referee Report · Anonymous (Referee 1) · 2023-6-17

Report

In the submitted manuscript, the authors develop a constructive approach to derive the convolution kernels in the spectral representation of generic multipoint correlation functions in the imaginary-frequency Matsubara formalism (MF).

Before this work, the precise form of the MF kernels was identified only up to three-point for bosonic correlation functions and up to four-point for fermionic correlation functions. A seminal work in this field (Ref. [21]) derived a generic formula of the "regular part" of the MF kernels, but that for the rest, called the "anomalous part", was not available.

This paper fills the gap by providing explicit derivation of the MF kernels for generic multipoint functions. Also, it's good to have a summary table (Table 1) that practitioners can easily look up.

In this regard, I strongly recommend the publication in SciPost Physics.
  • validity: -
  • significance: -
  • originality: -
  • clarity: -
  • formatting: -
  • grammar: -

Author:  Johannes Halbinger  on 2023-07-19  [id 3822]

(in reply to Report 1 on 2023-06-17)

We thank the referee for careful reading and the positive evaluation of the manuscript.

---

## Round 1 · Referee Report · Andrij Shvaika (Referee 2) · 2023-6-26

Strengths

See report.

Weaknesses

See report.

Report

In this article, the spectral representations of the Matsubara n-point correlation functions are derived. In the spirit of article by Kugler et al. [21], the Matsubara correlation function is expressed as a sum over all permutations of the system- and correlator-specific products of matrix elements multiplied by the universal kernel functions. An expressions for the arbitrary n-point kernel functions including anomalous terms are provided. In the second part of the article, the bosonic 3- and 4-point correlation functions for the fermionic Hubbard atom and a free spin of length S are considered. I have the following comments. 1. For the selected ordering of operators, the kernel function contains anomalous terms in the form of singularities when Ω_k^ab = i ω_k + E_a – E_b → 0. Only after summation over the cyclically related permutations, these singularities are mutually canceled [18,20], as well as many other terms including R_n, providing “additional” finite anomalous terms proportional to the powers of β depending on the order of singularities. Here, “additional” means with different spectral properties. In [21], in order to avoid (for unknown reasons) partial summation over cyclically related permutations, such finite anomalous terms were equally splitted between cyclically connected kernels without any justifications (see Appendix B in [21]). In this article, authors perform the same trick without any justifications too (see below). 2. It is not clear, how Eqs. (18) and (19) relate with the definition of h_k(τ_k) in (16). There are no any powers of τ_k in (16)! It looks like, authors are trying to obtain anomalous terms in powers of β, which will appear after final integration, by some tricks without performing summation over cyclically connected kernels. The statement “that we are only interested in the contribution K_n (z_n , z_n−1 , ..., z_2 ) that fulfills frequency conservation” is not enough and the detailed description of this important point is required. 3. Because of exploiting different tricks for obtaining kernel functions in this article and in [21], the resulting kernel functions are different and one have to perform summation over cyclically connected terms to show that the total Matsubara functions are equal (see Appendix A). That means that there are many contributions in the kernel functions unrequired for the final expressions. Are there any way within the proposed formalism to incorporate summation over cyclically connected terms from the beginning? Please, give some thoughts is it possible or not. Concluding, this article considers important problem of obtaining anomalous terms for the n-point Matsubara correlation functions and its results can be useful as for the analytical derivations, as for the numerical calculations, but it contains some unjustified tricks and assumptions which required detailed clarification. Only after that it can be considered for publication.

Requested changes

See report.

  • validity: ok
  • significance: high
  • originality: good
  • clarity: ok
  • formatting: reasonable
  • grammar: good

Author:  Johannes Halbinger  on 2023-07-19  [id 3823]

(in reply to Report 2 by Andrij Shvaika on 2023-06-26)
Category:
answer to question

We thank the referee for careful reading and the three questions raised in the report which we answer in the following.

1) To begin, we would like to point out the key difference between the pioneering and groundbreaking works in [18,20] and the approach chosen in [21] and in our manuscript: In References [18,20], the imaginary time integrals are performed to obtain the kernel functions and only afterwards the limits \Omega_k^ab = i \omega_k + E_a - E_b -> 0 are taken into account. To avoid possible singularities in the resulting denominators, a sum over cyclically related permutations has to be taken to obtain a finite result, giving rise to the anomalous terms. In our manuscript, however, we choose a different approach. Possible vanishing \Omega_k^ab are already taken into account on the level of evaluating the imaginary time integrals. Thus, our kernel functions are by definition free of any singularities and there is no need to sum over cyclically related permutations to cancel any singularities. The Kernel functions do NOT contain any singularities, for example see the K_2 of Eq. (30) in the revised manuscript. When it occurs that \Omega_1=0, the first term, - \Delta_1, evaluates to zero, see Eq. (29) and the sole contribution \beta/2 comes from the second ("anomalous") term. These terms proportional to \beta do not usually vanish after the permutations are summed in Eq. (15). In the contrary, they contain important physical information, for example encoding the Curie-susceptibility ~1/T=\beta in the free spin case, see the first line of Table 2. We are not aware of any unphysical information left in our final expression for the general Kernel functions in Eq. (26) and Table 1. For example, in the Hubbard atom example, the correlation functions (38)-(45) do contain anomalous terms of all orders.

2) First, we emphasize that for n-point correlation functions G, frequency conservation \Omega_1+\Omega_2+...+\Omega_n=0 (and z_1+z_2+...+z_n=0) is a fact that follows from the time translational invariance (4). It is no additional assumption that we invoke in the course of our calculation and a priori unrelated to the spectral representation! It is however a useful guiding principle that allows us to focus on the physically relevant part K_n of the more generally defined \mathcal{K}_n. The form of the ansatz for h_k(\tau_k) in (19) follows by inspection of the integrals in (16). Indeed, h_1(\tau_1)=exp(z_1 \tau_1), but if z_1=0 the integral over \tau_1 from 0 to \tau_2 yields \tau_2. This is the mechanism by which the powers of \tau_k can appear.

3) Before answering the question, we'd like to justify the approach of [21], especially Appendix B therein. The product of operator matrix elements, which are coined partial spectral functions in [21] and defined in (22b), fulfill the cyclicity relation (25), connecting cyclically related permutations. When considering anomalous contributions, where some of the energy differences vanish, the cyclicity relation implies that certain cyclically related partial spectral functions are identical. In turn, there is a certain degree of freedom of splitting anomalous contributions between exactly these permutations without affecting the final correlation function. In Appendix B, [21], the authors chose to split these anomalous contributions symmetrically. On the other hand, our approach boils down to splitting the same anomalous contributions asymmetrically. Most importantly, both approaches contain the same physical information distributed differently among the permutations, but there are no contributions in the kernel functions unrequired for the final expression. As mentioned in 1) above, we are not aware of any unphysical information left in our final expression for the Kernel functions in Eq. (26) and Table 1. Regarding Appendix A in our manuscript, we compare our 3rd order Kernel function (restricted to only one anomalous term) to the result of Ref. [21]. Our expression contains only 4 summands whereas the alternative expression contains 5 summands. We thus believe that our representation is even more compact. Additionally, we think that it is not straightforward to incorporate summation over cyclically connected terms in our approach. In fact, there is no need to do this in our approach as there are no singularities to be canceled, as pointed out in 1). Since our results are already very compact, we don't see an advantage in including these summations in our approach. In the contrary, we expect the results to become less compact.

We believe this answers the referee's question. We have carefully revised the manuscript for clarity and to avoid misconceptions, our changes appear in blue font.

---

## Round 2 · Referee Report · Anonymous (Referee 1) · 2023-7-23

Report

I again recommend the publication of this manuscript in SciPost Physics. I find the authors' reply to the second referee adequate.

---

## Round 2 · Referee Report · Anonymous (Referee 3) · 2023-7-30

Report

The authors have provided answers to all my comments and I recommend it for publication in SciPost Physics.

---

## Round 2 · List of Changes

Changes appear in blue font in the manuscript.

---

## Editorial Decision

published